# Anti-SARS-CoV-2S Antibody Levels in Healthcare Workers 10 Months after the Administration of Two BNT162b2 Vaccine Doses in View of Demographic Characteristic and Previous COVID-19 Infection

**DOI:** 10.3390/vaccines10050741

**Published:** 2022-05-09

**Authors:** Blanka Wolszczak-Biedrzycka, Anna Bieńkowska, Joanna Ewa Zaborowska, Elwira Smolińska-Fijołek, Grzegorz Biedrzycki, Justyna Dorf

**Affiliations:** 1Department of Psychology and Sociology of Health and Public Health, University of Warmia and Mazury in Olsztyn, 10-082 Olsztyn, Poland; anna.bienkowska@uwm.edu.pl; 2The Oncology Center of the Region of Warmia and Mazury in Olsztyn, Hospital of the Ministry of the Interior and Administration, 10-228 Olsztyn, Poland; 3District Hospital in Bartoszyce, 11-200 Bartoszyce, Poland; aniab446@wp.pl; 4Department of Physiology, Medical University in Gdańsk, 80-211 Gdańsk, Poland; elwira.smolinska@gumed.edu.pl; 5Hospital Dispensary, Regional Specialist Hospital in Olsztyn, 10-561 Olsztyn, Poland; gbiedrzycki@wss.olsztyn.pl; 6Department of Clinical Laboratory Diagnostics, Medical University of Białystok, Waszyngtona 15A, 15-269 Bialystok, Poland; justyna.dorf@umb.edu.pl

**Keywords:** BNT162b2, anti-SARS-CoV-2 antibodies, COVID-19, healthcare workers

## Abstract

Antibody levels that confer full protection against SARS-CoV-2 infection after the administration of different vaccine brands as well as the factors influencing the humoral immune response have been analyzed extensively ever since the vaccination program was launched in late 2020. The aim of this study was to determine anti-SARS-CoV-2S antibody titers in 100 healthcare workers 10 months after the administration of two BNT162b2 vaccine doses, and to investigate the influence of demographic characteristics, the presence of comorbidities and history of COVID-19 infection. The results were compared with antibody levels that were determined eight months after the administration of two BNT162b2 vaccine doses in our previous study. Antibody levels in venous blood serum were measured by the ECLIA method with the use of the Roche Cobas e411 analyzer. In all tested subjects, antibody titers remained high 10 months after vaccination, particularly in recovered COVID-19 patients, and only a minor decrease was observed relative to the values noted two months earlier.

## 1. Introduction

An effective vaccine against the SARS-CoV-2 virus was eagerly anticipated ever since the outbreak of the global COVID-19 pandemic. The Pfizer/BioNTech BNT162b2 mRNA vaccine was the first preparation approved for use in late 2020. The vaccine contains mRNA encoding the spike (S) protein that is found on the surface of the SARS-CoV-2 virus, which enables the S protein to bind to the ACE-2 receptor and penetrate the host cells [1]. The efficacy of the Pfizer/BioNTech vaccine against symptomatic infection with COVID-19 was initially estimated at 95% [1,2].

Both vaccines and natural infections elicit an immune response against the SARS-CoV-2 virus to protect the host organism. According to research, antibodies in the blood serum of recovered patients persist for at least three months after infection, and this period is much longer after vaccination [3,4]. Several studies have demonstrated that a history of COVID-19 infection combined with two vaccine doses induces the strongest humoral immune response [5,6]. The efficacy of mRNA vaccines against the Delta [4] and Omicron [5] variants of SARS-CoV-2 has also been analyzed.

Despite the fact that a full vaccination course provides considerable protection against infection, severe symptoms of COVID-19 and hospitalization, antibody levels decrease over time. New mutations of the virus have also emerged, and the global vaccination rate continues to be low. In addition, the factors influencing the effectiveness of vaccines have not been fully elucidated. It is well known that demographic characteristics (age, gender, BMI) and chronic diseases may affect the level of antibodies [7,8,9]. As a result, the risk of reinfection with SARS-CoV-2 is high, which is why a booster dose of the COVID-19 vaccine has been introduced in many countries around the world to increase protection against the virus [10].

The vaccination program can be optimized by monitoring changes in anti-SARS-CoV-2S antibody levels over time in different population groups to determine titers that confer full protection against infection, and to identify groups where humoral immune responses to vaccination may be weaker [11].

The aim of the present study was to assess the anti-SARS-CoV-2S antibody levels in healthcare workers 10 months after the administration of two BNT162b2 vaccine doses, compare the results with the values obtained two months earlier (8 months after the full vaccination course), and investigate the demographic characteristics, comorbidities and previous COVID-19 infection that may potentially affect antibody titers.

## 2. Materials and Methods

### 2.1. Materials

The population sample comprised 100 healthcare employees of the Hospital of the Ministry of Internal Affairs and Administration in Olsztyn. The analyzed subjects were aged 25 to 67 (mean 44, median 46), and the sample consisted of both males and females. The participants had been vaccinated with two doses of the BNT162B2 vaccine between 25 January and 17 February 2021. All participants gave their consent to participate in the study.

The studied population was divided into two groups: vaccinated individuals with a history of SARS-CoV-2 infection that had been confirmed by the GeneXpert PCR assay (based on the detection of two target genes: E and N2) before the administration of two vaccine doses (4Q 2020) (*n* = 50), and vaccinated individuals without a history of SARS-CoV-2 infection (*n* = 50).

Demographic data and information about comorbidities were collected from all participants. Each group was subdivided into subgroups based on gender (male, female), age (≤50, >50), type of work (medical, non-medical), BMI (≤24.9, >24.9), and presence of comorbidities in the past or at present, including diabetes, hypertension, cardiovascular diseases, autoimmune diseases, urinary tract diseases, hematological diseases and cancer (absent, present).

Blood for antibody tests was sampled twice: 8 months and 10 months after the administration of two BNT162B2 vaccine doses. The blood was collected into red-top Vacutainer tubes for serum separation by centrifugation. The blood was centrifuged for 10 min at 4000× *g* at room temperature. The serum was separated and frozen at −80 °C until analysis.

### 2.2. Method of Determining Antibody Levels

Anti-SARS-CoV-2-S antibody levels were measured in the electrochemiluminescence immunoassay (ECLIA) test with the Cobas e411 analyzer (Roche Diagnostics International Ltd, Rotkreuz, Switzerland), which enables the quantitative determination of antibodies (including IgG) to the SARS-CoV-2 spike (S) protein receptor binding domain (RBD) in human serum. The assay uses a recombinant protein representing the RBD of the S antigen in a double-antigen sandwich assay format, which favors the detection of high-affinity antibodies against SARS-CoV-2. The test is intended as an aid to assess the adaptive humoral immune response to the SARS-CoV-2 S protein. The detection threshold is ≥0.4 U/mL, and values ≥ 0.8 U/mL are considered positive. According to the manufacturer’s protocol, samples with a titer > 250 U/mL were diluted 100× at a time until the titer decreased to ≤250 U/mL [8].

### 2.3. Statistical Analysis

Data were processed statistically in GraphPad Prism 8.4.3 for Windows (GraphPad Software, La Jolla, CA, USA). The distribution of the results was analyzed using the Shapiro–Wilk test. A Student’s *t*-test was used for normal distribution, and the Mann–Whitney U test was applied to compare data that were not normally distributed. The results were presented as median values (minimum–maximum). Statistical significance was established at *p* < 0.05.

## 3. Results

### 3.1. Characteristics of the Study Group

The study included 100 vaccinated hospital workers. The population sample was divided into two groups: subjects with a history of COVID-19 (50 patients) and subjects without a history of COVID-19 (50 patients). In both groups, the majority of the participants were female (82%, 90%) and medical personnel (86%, 58%). Detailed characteristics of all groups are presented in Table 1. 

### 3.2. Comparison of Total Anti-SARS-CoV-2 Antibody Levels According to Sex, Age, BMI, Coexisting Diseases and Work Type in Group of Workers with History of COVID-19

We observed an increase in the difference in the total anti-SARS-CoV-2S antibody levels between the first (after 8 months) and second determination (after 10 months) in the group of men (change of 1.56%) compared to women (change of 1%) (*p* = 0.5814), as well as in the group of workers over 50 years old (change of 6.9%) compared to those younger than 50 years of age (change of 5%) (*p* = 0.1214) without a history of COVID-19 (Figure 1A,B), but these differences were not statistically significant. The total anti-SARS-CoV-2S antibody levels were also higher in the group of patients with increased BMI than in those with normal BMI (*p* = 0.6141) and in the group of medical workers in comparison with non-medical workers with a history of COVID-19 (*p* = 0.3123) (Figure 1C,E).

### 3.3. Comparison of Total Anti-SARS-CoV-2 Antibody Levels According to Sex, Age, BMI, Coexisting Diseases and Work Type in Group of Workers without History of COVID-19

We demonstrated a higher difference in the total anti-SARS-CoV-2S antibody levels between the first (after 8 months) and second determination (after 10 months) in the group of men than in the group of women without a history of COVID-19 (*p* = 0.1545) (Figure 2A). The total anti-SARS-CoV-2S antibody levels were increased in patients under 50 in comparison to those over 50 years old (*p* = 0.0497), as well as in the group of medical workers compared to non-medical workers with a history of COVID-19 (*p* = 0.0073) (Figure 2B,E).

### 3.4. Comparison of Total Anti-SARS-CoV-2 Antibody Levels between First (after 8 Months) and Second Determination (after 10 Months) According to Sex, Age, BMI, Coexisting Diseases and Work Type in Group of Workers with History of COVID-19

The total anti-SARS-CoV-2 antibody levels were significantly higher at 8 months than 10 months after vaccination in all compared groups, but the differences were not statistically significant (Figure 3).

### 3.5. Comparison of Total Anti-SARS-CoV-2 Antibody Levels between First (after 8 Months) and Second Determination (after 10 Months) According to Sex, Age, BMI, Coexisting Diseases and Work Type in Group of Workers with and without History of COVID-19

In all compared groups, the difference in the total anti-SARS-CoV-2 antibody levels between the first (after 8 months) and second determination (after 10 months) was higher in workers with a history of COVID-19 infection than in those without. In women, workers under and over 50 years old and in workers with normal and increased BMI, the differences between groups with and without a history of COVID-19 were statistically significant (*p* < 0.0001, *p* = 0.0292, *p* < 0.0001, *p* = 0.0031, and *p* = 0.0084, respectively). The difference in the total anti-SARS-CoV-2 antibody levels between the first (after 8 months) and second determination (after 10 months) was also greater in medical and non-medical workers and workers with and without coexisting diseases with a history of COVID-19 compared to those without (Figure 4).

### 3.6. Comparison of Total Anti-SARS-CoV-2 Antibody Levels after 10 Months on Second Determinations According to Sex, Age, BMI, Coexisting Diseases and Work Type in Group of Workers with and without History of COVID-19

The total anti-SARS-CoV-2 antibody levels after 10 months of the second dose of vaccination was significantly higher both in the group of women (*p* = 0.0102) and men (*p* = 0.0177) after COVID-19 compared to the patients without a history of COVID-19 (Figure 5A). We also demonstrated considerably higher levels of total anti-SARS-CoV-2 antibodies in the groups of patients under (*p* = 0.0005) and over 50 years old (*p* < 0.0001) with a history of COVID-19 in comparison to those without (Figure 5B). The total anti-SARS-CoV-2 antibody levels were significantly higher in the groups of patients with (*p* = 0.0005) and without (*p* < 0.0001) coexisting diseases with a history of COVID-19 in comparison to those without (Figure 5C). Medical and non-medical workers after COVID-19 had a considerably higher level of total anti-SARS-CoV-2 antibodies in comparison with workers without a history of COVID-19 (*p* = 0.0002, *p* = 0.0080) (Figure 5D). Patients with a history of COVID-19 both with normal (*p* = 0.0002) and higher BMI (*p* = 0.0001) had a significantly higher level of total anti-SARS-CoV-2 antibodies than in patients without (Figure 5E).

## 4. Discussion

Antibody titers that confer full protection against COVID-19 infection after vaccination and the rate at which antibody levels decrease over time in different population groups have not been unambiguously established to date. For this reason, numerous research studies have been undertaken to analyze anti-SARS-CoV-2S antibody levels in response to different vaccine brands, to determine changes in antibody levels within different periods of time after the first and the second vaccine dose, and to evaluate the influence of demographic characteristics, comorbidities and previous COVID-19 infection on antibody titers.

In our previous study, healthcare workers were examined eight months after the administration of two doses of the BNT162B2 vaccine. The study revealed that the anti-SARS-CoV-2S antibody levels were significantly higher in healthcare workers with a history of COVID-19 infection (2162 U/mL) than in subjects who had not been infected (522 U/mL) [9]. The present study was undertaken two months later, i.e., 10 months after the administration of two doses of the BNT162B2 vaccine, and it involved the same subjects. In both groups, the antibody levels were still significantly higher than the cut-off value in the Elecsys SARS-CoV-2 antigen assay (0.4 U/mL), which points to a strong humoral immune response. Median anti-SARS-CoV-2 antibody levels were determined at 1687 U/mL in convalescent individuals and at 449 U/mL in subjects without a history of infection. What is more, the antibody levels were higher in each group of workers with history of COVID-19 according to gender, age, BMI, type of workers and coexisting diseases. The present results corroborate the findings of other authors, which confirms that vaccination produces a much stronger humoral immune response in patients with a history of COVID-19 [5,12,13,14,15].

The rate of changes in antibody levels over a period of two months (8 vs. 10 months after the administration of two vaccine doses) was analyzed in detail. Antibody titers decreased in both groups: by 13% in subjects without a history of infection, and by 21% in recovered COVID-19 patients. When differences in antibody levels between the two sampling dates were adjusted for age, gender, type of work, BMI and comorbidities, a greater decrease (*p* < 0.05) was noted in healthcare workers who had recovered from COVID-19, but their antibody levels were still significantly higher than in individuals without a history of infection. This observation could be attributed to the fact that the higher the antibody titer at baseline, the greater the decrease over time. Terpos et al. [13] analyzed the rate of a decrease in antibody levels over a period of three months, i.e., six and nine months after the administration of two BNT162B2 doses. The anti-SARS-CoV-2S antibody levels decreased from 523.8 U/mL to 367.1 U/mL on average (by approx. 30%) in the investigated period. 

The immune response to vaccine brands other than Pfizer/BioNTech was also examined in healthcare workers. Tré-Hardy et al. [16] analyzed the immunogenicity of the Moderna mRNA-1273 COVID-19 vaccine and found that antibody titers were higher in subjects who had recovered from COVID-19 than in individuals without a history of infection six months after the full vaccination course (two doses). The cited authors also reported a significant decrease in antibody levels three to six months after the second dose. In a study by Doria-Rosa et al. [17], antibodies were detected six months after the second dose of the mRNA-1273 vaccine in persons without a history of SARS-CoV-2 infection.

Higher antibody levels in recovered COVID-19 patients have been investigated by numerous authors. In previously infected subjects, antibody titers were already relatively high after the first dose of the BNT162b2 vaccine, and they were often higher than in fully vaccinated persons without a history of infection. Moreover, a second dose did not lead to a significant increase in antibody levels in recovered COVID-19 patients [14,18]. This observation should be taken into account to determine the optimal timing for a booster dose in the previously infected population. However, each case should be examined individually based on the humoral and cellular immune responses.

As previously mentioned, antibody titers that confer full protection against COVID-19 have not been unambiguously established. However, valuable observations have been made by several researchers. According to Dimeglio et al. [19], vaccine-induced antibody titers of 141–1700 U/mL provide 89.3% protection against infection, whereas titers higher than 1700 U/m confer full protection. In the current study, such high antibody levels were found in around 70% of recovered healthcare workers, but they were not noted in any of the previously uninfected subjects 10 months after the full vaccination course. The above observation can be helpful in establishing the minimum number of antibody titers for receiving a booster dose. However, long-term multi-center observations are needed to confirm the hypothesis that previously infected individuals do not require a booster dose or that a booster dose can be administered to recovered patients at a later date than to subjects without a history of COVID-19.

It is difficult to identify factors other than a previous SARS-CoV-2 infection that condition higher antibody levels. Some researchers found that seniors (>60) produced fewer antibodies than younger subjects (<60) [20,21], whereas age-related differences were not observed in other studies [22,23]. In the present study, anti-SARS-CoV-2S antibody levels were higher in 50+ healthcare workers with and without a history of COVID-19 than in subjects younger than 50. Differences in antibody titers between genders and between individuals with different BMIs have also been reported in the literature. Some research suggests that women produce relatively more antibodies than men [18,19,20]. This observation was confirmed in the present study, but only in subjects without a history of COVID-19. However, when the results were adjusted for BMI, persons with a BMI > 24.9 produced a stronger humoral immune response than individuals with a BMI < 24.9. No such correlations were reported by other researchers [18,23].

Healthcare workers are at a particularly high risk of becoming infected with COVID-19, which is why all reports that could help optimize the vaccination strategy in this population group, including the administration of a booster dose, are highly valuable. We should always remember that this profession is particularly exposed to frequent and long-lasting viral infections and that there are still new mutations for which the available vaccines may not be fully prepared. However, there are no clear guidelines regarding protective titers, and it should be stressed that even high antibody levels in this population group do not guarantee full protection against COVID-19, therefore a booster dose should be recommended until the results of long-term observations are known. The main limitations of this study were that it was conducted in a single medical facility and involved a relatively small sample. Despite the above, the obtained results can significantly contribute to improving the vaccination program for healthcare employees.

## 5. Conclusions

This study demonstrated that the anti-SARS-CoV-2S antibody levels in healthcare workers remained relatively high 10 months after the administration of two doses of the BNT162b2 vaccine, particularly in subjects with a history of COVID-19, and only a minor decrease was observed relative to the values noted two months earlier.

## Figures and Tables

**Figure 1 vaccines-10-00741-f001:**
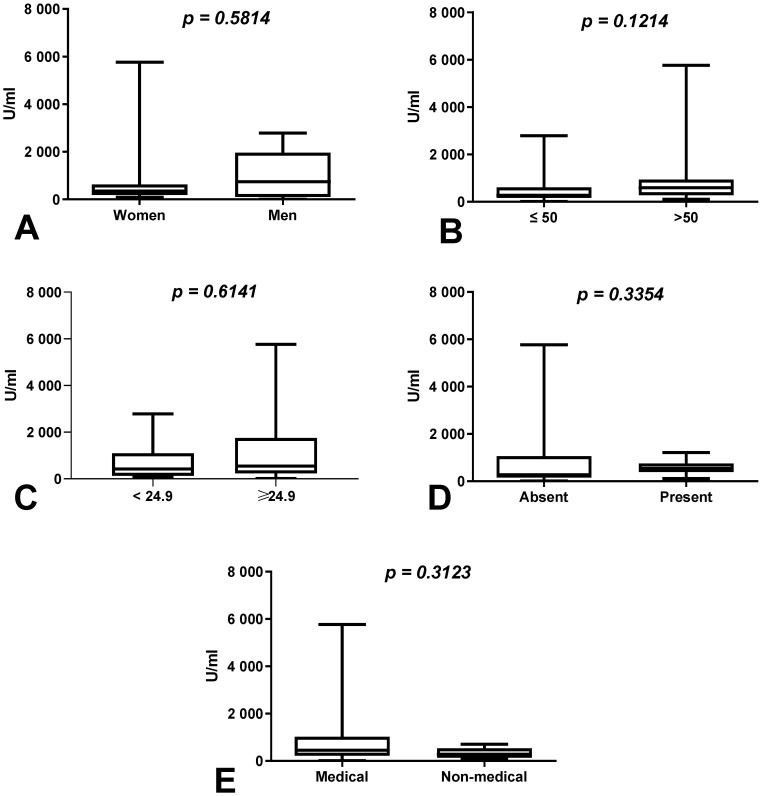
(**A**) Comparison of difference in total anti-SARS-CoV-2 antibody levels between first (after 8 months) and second determination (after 10 months) in the groups of women and men with a history of COVID-19. The data are presented as median (minimum–maximum). (**B**) Comparison of difference in total anti-SARS-CoV-2 antibody levels between first and second determination in the groups of patients over and under the age of 50 with a history of COVID-19. The data are presented as median (minimum–maximum). (**C**) Comparison of difference in total anti-SARS-CoV-2 antibody levels between first and second determination in the group of patients with normal and increased BMI with a history of COVID-19. The data are presented as median (minimum–maximum). (**D**) Comparison of difference in total anti-SARS-CoV-2 antibody levels between first and second determination in the groups of patients with and without coexisting diseases with a history of COVID-19. The data are presented as median (minimum–maximum). (**E**) Comparison of difference in total anti-SARS-CoV-2 antibody levels between first and second determination in the groups of medical and non-medical workers with a history of COVID-19. The data are presented as median (minimum–maximum).

**Figure 2 vaccines-10-00741-f002:**
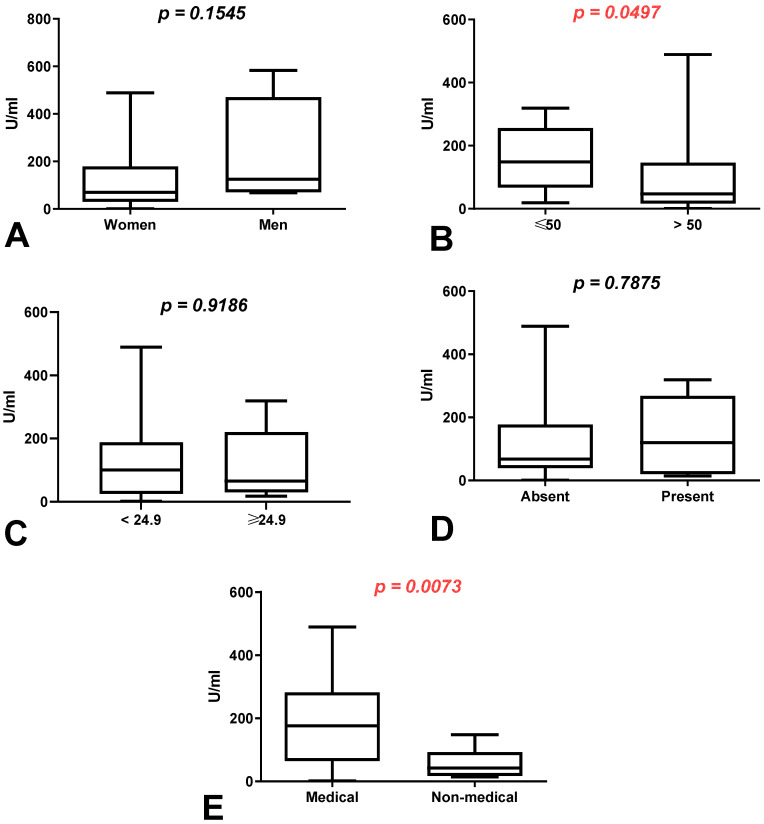
(**A**) Comparison of difference in total anti-SARS-CoV-2 antibody levels between first (after 8 months) and second determination (after 10 months) in the groups of women and men without a history of COVID-19. The data are presented as median (minimum–maximum). (**B**) Comparison of difference in total anti-SARS-CoV-2 antibody levels between first (after 8 months) and second determination (after 10 months) in the groups of patients under and over the age of 50 without a history of COVID-19. The data are presented as median (minimum–maximum). (**C**) Comparison of difference in total anti-SARS-CoV-2 antibody levels between first (after 8 months) and second determination (after 10 months) in the group of patients with normal and increased BMI without a history of COVID-19. The data are presented as median (minimum–maximum). (**D**) Comparison of difference in total anti-SARS-CoV-2 antibody levels between first (after 8 months) and second determination (after 10 months) in the groups of patients with and without coexisting diseases without a history of COVID-19. The data are presented as median (minimum–maximum). (**E**) Comparison of difference in total anti-SARS-CoV-2 antibody levels between first (after 8 months) and second determination (after 10 months) in the groups of medical and non-medical workers without a history of COVID-19. The data are presented as median (minimum–maximum).

**Figure 3 vaccines-10-00741-f003:**
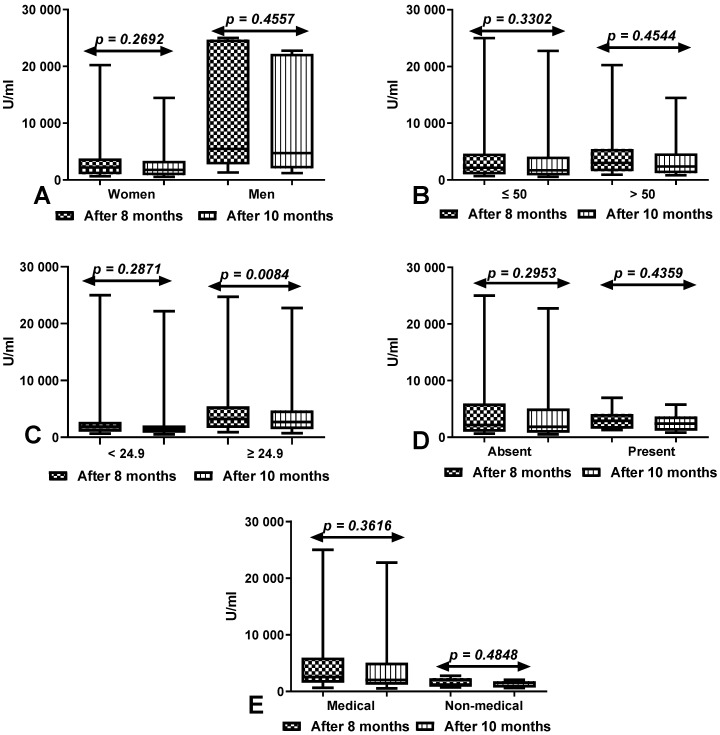
(**A**) Comparison of total anti-SARS-CoV-2 antibody levels between first (after 8 months) and second determination (after 10 months) in the groups of women and men with a history of COVID-19. The data are presented as median (minimum–maximum). (**B**) Comparison of total anti-SARS-CoV-2 antibody levels between first (after 8 months) and second determination (after 10 months) in the groups of patients under and over the age of 50 with a history of COVID-19. The data are presented as median (minimum–maximum). (**C**) Comparison of total anti-SARS-CoV-2 antibody levels between first (after 8 months) and second determination (after 10 months) in the groups of patients with normal and increased BMI with a history of COVID-19. The data are presented as median (minimum–maximum). (**D**) Comparison of total anti-SARS-CoV-2 antibody levels between first (after 8 months) and second determination (after 10 months) in the groups of patients with and without coexisting diseases with a history of COVID-19. The data are presented as median (minimum–maximum). (**E**) Comparison of total anti-SARS-CoV-2 antibody levels between first (after 8 months) and second determination (after 10 months) in the groups of medical and non-medical workers with a history of COVID-19. The data are presented as median (minimum–maximum).

**Figure 4 vaccines-10-00741-f004:**
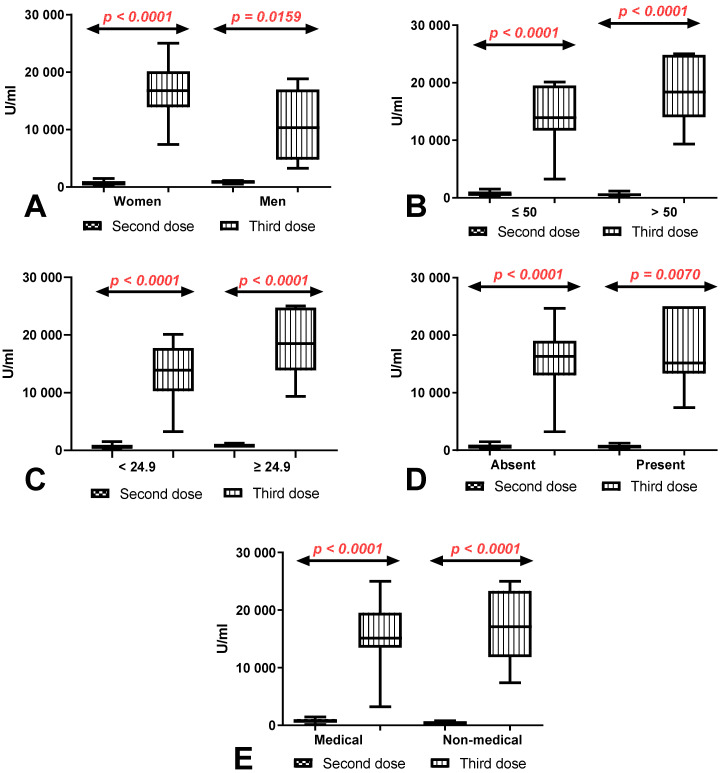
(**A**) Comparison of difference in total anti-SARS-CoV-2 antibody levels between first (after 8 months) and second determination (after 10 months) in the groups of women and men with and without a history of COVID-19. The data are presented as median (minimum–maximum). (**B**) Comparison of difference in total anti-SARS-CoV-2 antibody levels between first (after 8 months) and second determination (after 10 months) in the groups of patients under and over the age of 50 with and without a history of COVID-19. The data are presented as median (minimum–maximum). (**C**) Comparison of difference in total anti-SARS-CoV-2 antibody levels between first (after 8 months) and second determination (after 10 months) in the group of patients with normal and increased BMI with and without a history of COVID-19. The data are presented as median (minimum–maximum). (**D**) Comparison of difference in total anti-SARS-CoV-2 antibody levels between first (after 8 months) and second determination (after 10 months) in the groups of patients with and without coexisting diseases with and without a history of COVID-19. The data are presented as median (minimum–maximum). (**E**) Comparison of difference in total anti-SARS-CoV-2 antibody levels between first (after 8 months) and second determination (after 10 months) in the groups of medical and non-medical workers with and without a history of COVID-19. The data are presented as median (minimum–maximum).

**Figure 5 vaccines-10-00741-f005:**
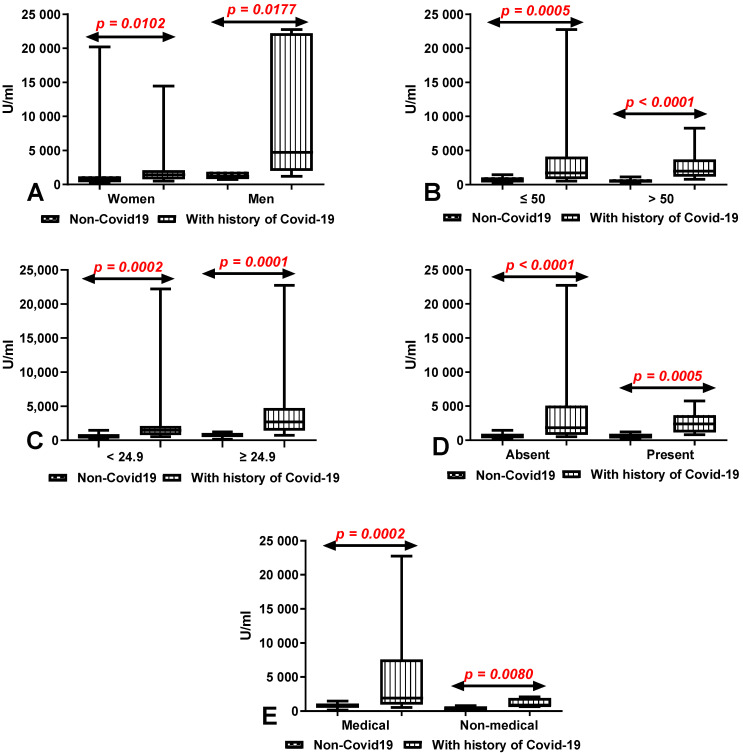
(**A**) Comparison of total anti-SARS-CoV-2 antibody levels in groups of women and men with and without a history of COVID-19. The data are presented as median (minimum-maximum). (**B**) Comparison of total anti-SARS-CoV-2 antibody levels in groups of patients under and over the age of 50 with and without a history of COVID-19. The data are presented as median (minimum-maximum). (**C**) Comparison of total anti-SARS-CoV-2 antibody levels in groups of patients with and without coexisting diseases with and without a history of COVID-19. The data are presented as median (minimum-maximum). (**D**) Comparison of total anti-SARS-CoV-2 antibody levels in groups of medical and non-medical workers with and without a history of COVID-19. The data are presented as median (minimum-maximum). (**E**) Comparison of total anti-SARS-CoV-2 antibody levels between group of patients with normal and increased BMI with and without a history of COVID-19. The data are presented as median (minimum–maximum).

**Table 1 vaccines-10-00741-t001:** Characteristic of study groups.

Workers with History of COVID-19 (*n* = 50)															
	**Level of Anti-SARS-CoV2S Level (U/mL)) after 8 months**								**Level of Anti-SARS-CoV2S Level (U/mL)) after 10 months**						
**Parameter**	**Number of workers (%)**	**Median**	**Mean**	**SD**	**Q1**	**Q3**	**5 percentile**	**95 percentile**	**Median**	**Mean**	**SD**	**Q1**	**Q3**	**5 percentile**	**95 percentile**
**Age**															
≤50	31 (62%)	1497	2304	2203	877.5	3001	456.9	8293	1324	9586	8965	615	4110	538.4	22,710
>50	19 (38%)	2564	3210	2324	1624	3751	901	10077	2387	3569	4235	1185	4670	800	14,466
**Sex**															
female	41 (82%)	1705	2914	3575	983	3457	516.5	10461	1687	2697	3159	829	3365	578.3	12,064
male	9 (18%)	4603	8238	9597	2046	15605	991	25000	4531	8639	9584	2030	2105	1210	22,765
**Type of workers**															
medical	43 (86%)	2297	4371	5816	1304	4299	681.1	24051	2030	4541	5837	1214	5087	629.5	22,489
non-medical	7 (14%)	941.5	1255	852.2	565	2028	398	2794	857	1187	588	721	1813	641	2081
**BMI**															
≤24.9	18 (36%)	1850	3926	6195	982.5	2719	661	25000	1471	3328	5520	809	2091	527	22,212
>24.9	32 (64%)	3244	5390	6460	1665	5455	901	24726	2693	4503	5490	1452	4731	732	22,765
**Other diseases**															
present	23 (46%)	2901	3215	1775	1543	4110	1320	6987	2497	2637	1564	1080	3843	561.2	22,599
absent	27 (54%)	2190	5323	7318	996	5970	686.2	24,918	2037	4497	6341	1010	3233	836	5779
**Workers without History of COVID-19 (n = 50)**															
**Level of Anti-SARS-CoV2S Level (U/mL)) after 8 Months**									**Level of Anti-SARS-CoV2S Level (U/mL)) after 10 Months**						
**Parameter**	**Number of workers (%)**	**Median**	**Mean**	**SD**	**Q1**	**Q3**	**5 percentile**	**95 percentile**	**Median**	**Mean**	**SD**	**Q1**	**Q3**	**5 percentile**	**95 percentile**
**Age**															
≤50	24 (48%)	507.5	629.3	419.7	300.3	372.8	189.8	1707	476.5	628	320	246	1070	198	1476
>50	26 (52%)	522.5	673.8	476.1	372.8	770.8	372.8	770.8	514.5	485	289	327	786	159	1129
**Sex**															
female	45 (90%)	778.5	1675	3501	497.3	1547	222.2	9571	496	625	370	341	1005	166.8	1430
male	5 (10%)	1222	1119	428.1	653	1512	512	1564	457	587	243	424	943	478	1100
**Type of workers**															
medical	29 (58%)	626.0	803.4	512.3	433.5	1222	210	1891	609	685	364	493	1094	159	1476
non-medical	21 (42%)	453	443.9	202.4	257.5	552.5	145.7	791.7	359	415	384	294	534	198	812
**BMI**															
≤24.9	29 (58%)	535	726.5	467	382	1048	217	1755	496	600	367	330	908	198	1476
>24.9	21 (42%)	583	750.4	451.1	510	1142	225	1564	539	797	309	491	1123	485	1245
**Other diseases**															
present	20 (40%)	628.5	743.3	472.8	375	1185	235	1564	530	604	364	307	939	221	1245
absent	30 (60%)	569	728.6	458.1	434.5	1055	217	1755	516	644	375	356	956	159	1476

## Data Availability

The full data presented in this study are available on request from the corresponding author.

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
