# Peer review of "Anti-SARS-CoV-2S Antibody Levels in Healthcare Workers 10 Months after the Administration of Two BNT162b2 Vaccine Doses in View of Demographic Characteristic and Previous COVID-19 Infection"

_vaccines, 2022, doi:10.3390/vaccines10050741_

Round 1

Reviewer 1 Report

The manuscript is well presented and written, even if more attention should be paid to English grammar and structure.

Despite a good presentation, the manuscript in this form is not technically sound. I suggest refining the aims and uniforming them from the abstract to the introduction. Additionally, I really cannot imagine why demographic factors and the presence of comorbidities could affect the antibody levels. This should be better clarified in the introduction. I recommend only focusing on what this study adds to the previous literature for the conclusions. In this sense, why it is important a booster dose when you demonstrated high levels of antibodies after 10 months to the second dose? All these aspects should be clarified and defended in the discussion and deleted from the conclusions. 

Reviewer 2 Report

Comments:

Overall the article is of interest and has a good potential buti t needs to be improved.

  1. Line 68: mean and median age of patients should be stated.
  2. The studied population seems well balanced, though testing for statistical homogeneity of gender, age, comorbidities, and possibly also BMI, among the two main studied subgoups (only vaccined individuals vs individuals who a history of SARS-CoV-2 infection) is needed.
  3. Line 83 you mean after “the second “vaccine dose?
  4. Line 114-121 As the authots correctly stated, the observed differences were not statistically significant. Still it coulld be of interest to report some percentages even though, statistically not significant.

  1. RESULTS:

The authors compare total anti-SARS-CoV-2 antibodies levels after 8 and 10 months from vaccination in different subgroups (based on sex, age, BMI, coexisting diseases and work type) among individuals with a history of COVID 19 and vaccination (Results 3.2).

Separately they then compare total anti-SARS-CoV-2 antibodies levels after 8 and 10 months from vaccination in different subgroups (based on sex, age, BMI, coexisting diseases and work type) in individuals without history of COVID 19 and vaccination (Results 3.3).

Though these informations are of interest, a comparison of total anti-SARS-CoV-2 antibodies levels in the main groups (individuals with a history of COVID 19 and vaccination vs. individuals without history of COVID 19 and vaccination) at 10 months from the second vaccine dose (possibly also stratified for sex, age, BMI, coexisting diseases and work type) would be of greater interest and is actually needed to state line 167-169.

  1. DISCUSSION section (l. 167-169):

Indeed the authors state in the DISCUSSION section (l. 167-169) that their “results corroborate the findings of other authors, which confirms that vaccination produces a much stronger humoral immune response in patients with a history of COVID-19 [5, 13, 14].”

This can be stated however only if the authors statistically compare the two main groups (individuals with a history of COVID 19 and vaccination vs. individuals without history of COVID 19 and vaccination) at 10 months.

Round 2

Reviewer 1 Report

The manuscript significantly improved after the revision. I have some concerns regarding the Authors' reply. They wasted time and words explaining things that are obvious (i.e., point 2 and its related answer), but I asked to be more clear, concise, and direct. In fact, the Authors confuse demographic factors (Ethnicity, Geographic Area, Educational attainment, Income level) with demographic characteristics (age, gender, BMI). So, the concerns are still unsolved. Before any acceptance for publication here or elsewhere, I recommend to carefully read all the previous comments and address them adequately, without arrogance. Finally, there are missing the standard deviation and the Q1, Q3 of the mean and median of the population. 
